# Non-Invasive Brain Stimulation for Children with Autism Spectrum Disorders: A Short-Term Outcome Study

**DOI:** 10.3390/bs7030063

**Published:** 2017-09-17

**Authors:** Lázaro Gómez, Belkis Vidal, Carlos Maragoto, Lilia Maria Morales, Sheyla Berrillo, Héctor Vera Cuesta, Margarita Baez, Marlén Denis, Tairí Marín, Yaumara Cabrera, Abel Sánchez, Celia Alarcón, Maribel Selguera, Yaima Llanez, Lucila Dieguez, María Robinson

**Affiliations:** 1Non Invasive Brain Stimulation Unit, International Center for Neurological Restoration, 25th Ave, Playa, Havana 15805, Cuba; maragoto@neuro.ciren.cu (C.M.); verac@neuro.ciren.cu (H.V.C.); mdenis@neuro.ciren.cu (M.D.); tairi@neuro.ciren.cu (T.M.); yaumara@neuro.ciren.cu (Y.C.); abel@neuro.ciren.cu (A.S.); 2Child and Adolescent Mental Health Service, Borrás-Marfán Hospital, G and 27 Street., Vedado, Havana 10400, Cuba; bvidalm@infomed.sld.cu (B.V.); marser@infomed.sld.cu (M.S.); 3Neuropediatric Clinic, International Center for Neurological Restoration, 25th Ave. Playa, Havana 15805, Cuba; yllanez@neuro.ciren.cu; 4Clinical Neurophysiology Lab., International Center for Neurological Restoration, 25th Ave. Playa, Havana 15805, Cuba; lily@neuro.ciren.cu (L.M.M.); sheyla@neuro.ciren.cu (S.B.); minou@neuro.ciren.cu (M.B.); calarcon@neuro.ciren.cu (C.A.); lucila@neuro.ciren.cu (L.D.); 5Clinical Immunology Lab., International Center for Neurological Restoration, 25th Ave. Playa, Havana 15805, Cuba; robin@neuro.ciren.cu

**Keywords:** autism, transcranial direct current stimulation, repetitive transcranial magnetic stimulation

## Abstract

Non-Invasive Brain Stimulation (NIBS) is a relatively new therapeutic approach that has shown beneficial effects in Autism Spectrum Disorder (ASD). One question to be answered is how enduring its neuromodulatory effect could be. Twenty-four patients with ASD (mean age: 12.2 years) received 20 sessions of NIBS over the left dorsolateral prefrontal cortex (L-DLPFC). They were randomized into two groups with two (G1) or three (G2) clinical evaluations before NIBS. Both groups had a complete follow-up at six months after the intervention, with the aim of determining the short-term outcome using the total score on the Autism Behavior Checklist, Autism Treatment Evaluation Checklist, and the Autism Diagnostic Interview. Transcranial Direct Current Stimulation (tDCS) was used in ASD patients aged <11 years, and repetitive Transcranial Magnetic Stimulation (rTMS) for 11–13-year-olds. Observation points were at one, three, and six months after completing all the sessions of NIBS. A significant reduction in the total score on the three clinical scales was observed and maintained during the first six months after treatment, with a slight and non-significant tendency to increase the scores in the last evaluation. Twenty sessions of NIBS over the L-DLPFC improves autistic symptoms in ASD children, with a lasting effect of six months.

## 1. Introduction

Autism Spectrum Disorders (ASD) still pose challenges for neuroscience for many reasons; one of these is how to apply the advances in neuroscience to understanding and treating ASD by improving diagnosis, interventions, and treatments. To date, diagnosis is made on clinical bases in the absence of a true biomarker for diagnosis and a better therapeutic approach. Different pharmacological and non-pharmacological therapies have shown positive results but are far from leading to a significant improvement in the core symptoms of patients with ASD. A high incidence of brain anomalies had been described by different methods in autistic patients in comparison with neurotypical patients, including MRI techniques and anatomic studies [1,2,3]. One interesting feature described by Casanova et al. was the abnormal structure in cortical mini-columns, with poor neuropil development in ASD patients, which probably accounts for intracortical inhibitory dysfunction [1,4]. Alterations in the GABAergic signaling pathway may characterize autistic neurobiology, which seems to not simply be related to a decreased GABA concentration, but probably to perturbations in key components of the GABAergic pathway beyond GABA levels, such as receptors and inhibitory neuronal density [5].

There is also some research exploring functional intracortical inhibition in motor areas of ASD patients that reinforce these ideas; nevertheless, its functional expression in terms of modulation by means of paired pulse stimulation protocol with TMS is really less remarkable than was expected, and it has been demonstrated in only a fraction of the ASD population [6]. In other experiments, indirect evidence has been described related to an aberrant plasticity based on an imbalanced excitatory and inhibitory cortical tone as well as biased GABAergic dysfunction [7].

Casanova et al. were the first to propose the use of Non-Invasive Brain Stimulation (NIBS) for ASD patients, based on the possibility that low-frequency repetitive TMS could somehow improve GABAergic neurotransmission; in their case, low-frequency repetitive Transcranial Magnetic Stimulation (rTMS) was used as a way to improve intracortical inhibition in ASD patients. They applied one weekly session over 12 weeks in a group of 25 ASD patients and described an increase in gamma activity in the EEG evoked by a visual processing paradigm, with changes in Event Related Potential (ERP), improved error monitoring, and correction function in a visual recognition task after the intervention [8,9]. There are just a few other studies providing evidence of the potential efficacy of both rTMS and transcranial direct current stimulation (tDCS) in ASD (excluding case reports). Unfortunately, most of the trials are limited to a low number of sessions (not more than 15 in the best-case scenario), and it was not known how enduring the modulatory effect was [10,11,12,13].

Dorsolateral prefrontal cortex (DLPFC) is the preferred target for many other clinical conditions, thus also in ASD based on its connections with other cortical and subcortical structures. Nevertheless, other considerations are valid about the feasibility for the selection of other targets [1,14]. Studies in ASD patients using low-frequency rTMS over the left DLPFC (L-DLPFC) have shown a significant improvement in some components of event-related cortical potentials (N200 and P300) as well as a significant reduction in response errors during cognitive tasks, repetitive behavior, and irritability [9]. Other authors using cathodal tDCS over the L-DLPFC have also reported beneficial effects for ASD in young adults. Curiously, there is also a published work of a trial with anodal stimulation over DLPFC reporting improvement in autistic symptomatology [10,11].

If we accept the hypothesis that an intra-cortical inhibitory dysfunction exists in ASD patients that can contribute to several dysfunctions in the motor, sensorial, and cognitive domains, we can expect some improvement in autistic behavior after applying brain-stimulating protocols that may potentiate intracortical inhibition, probably increasing the amount of available endogenous GABA in neuronal networks related to the stimulation target. In previous experiments, low-frequency rTMS and cathodal stimulation over the left DLPFC in children with ASD were associated with an improvement in autistic behavior one week after completing 20 sessions of transcranial stimulation [15,16]. One important question is how lasting this effect would be; in the present study, we describe the short-term clinical response during the first six months after treatment. We hypothesized that 20 sessions of NIBS over the L-DLPFC are expected to have a lasting positive neuromodulatory effect in patients with ASD.

## 2. Materials and Methods

### 2.1. Study Design

A controlled, randomized, and partial crossover trial was carried out in 24 children with ASD. Fifteen patients received the intervention after two clinical evaluations within one month (group 1, G1); nine patients started receiving the intervention after two months with three evaluation points at one-month intervals (group 2, G2). We assume that according to the stimulation protocol to be used (one daily session, up to a total of 20), it would be difficult to determine a washout period and patients from G2 ran in parallel to patients from G1 while they were receiving the intervention.

### 2.2. Sample Selection and Group Distribution

Children with ASD were recruited from the ambulatory service of the International Center for Neurological Restoration and Marfán-Borrás Hospital, from November 2015 to October 2016. Their diagnosis was made based on DSM-V diagnostic criteria, with the consensus of a multidisciplinary team including two Child Psychiatrists and two Neuropediatric Specialists. Only patients with a slight or moderate grade of severity were included according to their clinical characteristics, with stability during the last two months, and no changes in their therapeutic scheme in either pharmacologic or non-pharmacologic interventions. Diagnosis was confirmed by the results of the Childhood Autism Rating Scale (CARS) [17] and the Autism Diagnostic Interview, Revisited Edition (ADI-R, diagnostic algorithm) [18].

Patients with severe autistic behavior were excluded, as well as patients with concurrent epilepsy. Patients with unclear diagnosis or with diagnostic disagreement between neurologist and psychiatrist opinion were also excluded. If any change was needed in their therapeutic scheme, the patient was also excluded from the trial. We applied a simple randomization technique to assign patients to G1 (early intervention group, after one month of follow-up) or G2 (intervention after two months of follow-up).

### 2.3. Clinical Evaluation

Three main clinical scales were used as outcome measurements for all patients, according to their parent opinions: ADI-R (algorithm for current condition), the Autism Behavioral Checklist (ABC) [19], and the Autism Treatment Evaluation Checklist (ATEC) [20]. All the scales were applied by a Child Psychiatrist who was not involved in the intervention, and patients were evaluated at one, three, and six months after completing the 20 sessions. The Global Clinical Impression Scale (GCIS) [21] was also applied as a complementary, more qualitative evaluation. The Wilcoxon matched pair test and Mann–Whitney U test were used either for intra- or between-group comparisons at the different evaluation moments (α = 0.05).

### 2.4. Neurophysiological Evaluation

#### 2.4.1. Functional Brain Connectivity

An electroencephalogram (EEG) based connectivity analysis was carried out in all the ASD patients, but only 15 good-quality EEG recording were obtained for connectivity analysis (nine from the tDCS group and six from the rTMS group). EEG traces taken one week before starting the intervention and one week after finishing it were analyzed. Significant windows were selected from the EEG trace in open eyes state (38 from each patient), because they were more often artifact-free than the closed-eyes state.

We used a 19-electrode montage including Fp1, Fp2, F7, F8, F3, F4, C3, C4, T5, T6, T3, T4, P3, P4, O1, O2, Fz, Cz, and Pz, from the 10/20 system. Electrode Impedance was kept below 5 kΩ. Functional connectivity was analyzed based on the synchronization likelihood between electrodes for the five frequency bands: δ (1–3.9 Hz), θ (4–7.9 Hz) α (8–12.9 Hz), β (13–29.9 Hz), and γ (30–35 Hz) [22]. Mathematical analysis was developed with connectivity algorithms implemented in MATLAB v.7.7 R2008b. Significant connectivity (*p* < 0.05) in each frequency band was represented on an X/Y coordinate according to the 10/20 system for each brain stimulation modality independently, and the results by group were represented over the scalp surface map.

#### 2.4.2. Event-Related Potentials

Only in six children was it possible to carry out a passive oddball paradigm for P300 Event-Related Potentials (ERPs) before the intervention; four of them received tDCS and two rTMS. Patients were seated in a sound- and light-attenuated room while a paradigm was conducted consisting of 200 stimuli, 80% frequent (500 Hz) and 20% infrequent target (1000 Hz) tones, delivered through headphones while children were watching a silent movie. In the passive version of the oddball task the subject´s attention is usually directed away from the sequence of standard and deviant tones toward another, moderately demanding task, usually in a different modality [23]. We proposed a version of this paradigm considering the characteristics of the autistic children, with poor collaboration. All stimuli (50 ms; 5 ms rise and fall time) were presented binaurally with an inter-stimulus interval of 1300 ms.

To record the auditory P300 ERPs, 19 surface electrodes (Ag/ClAg) were attached to the scalp according to the international electrode placement 10–20 system (Fp1, Fp2, F3, F4, F7, F8, C3, C4, T3, T4, T5, T6, P3, P4, O1, O2, Fz, Cz, Pz), with additional electrodes for EOG (lateral to the outer canthus of the right eye and above the middle of the left eyebrow), and referenced to the linked earlobe. The electrode impedance was kept below 5 kΩ. EEG data were off-line averaged (100 ms prior and 700 ms after stimulus onset), and P300 latency and amplitude were quantified at Fz, Cz, and Pz. A continuous acquisition system was employed (Medicid 5, Neuronic SA, Cuba) and EEG data were EOG-corrected offline. The sampling rate of all channels was 200 Hz. To assure auditory system normality, Auditory Brainstem Response was previously recorded. The stimuli were 0.1 ms alternating clicks delivered through a headphone (DR-531B-7, Elegas Acous Co. Ltd., Tokyo, Japan). The records were obtained using the evoked potentials measuring system Neuropack M1 (Nihon Kohden, Tokyo, Japan).

P300 ERPs were obtained before and after NIBS, and P300 latency and amplitude values from the two evaluations were compared (from Fz electrode position; Wilcoxon matched pairs test, α < 0.05). All measurements were of the difference wave (infrequent wave minus frequent wave) and in this case, due to the low number of patients’ recordable P300, the analysis was performed considering all patients as a single group.

### 2.5. Intervention

Patients received one daily session of NIBS, from Monday to Friday, for a total of 20 sessions. tDCS was used in patients 10 years old or younger, and rTMS in patients 11 and older. The reason for using tDCS in younger children instead of rTMS was that in this case a major degree of collaboration is needed to assure effective focal stimulation over the selected target (left dorsolateral prefrontal cortex, L-DLPFC). During the stimulation sessions patients, were comfortably seated while watching TV cartoons of their preference or listening to music and playing with small, simple toys.

#### 2.5.1. Transcranial Direct Current Stimulation (tDCS)

tDCS was only used in patients 10 years old or younger (Neuroconn tDCS stimulator, München, Germany). A cathode was positioned over F3 (10/20 international EEG electrode system), and an anode over the proximal right arm. The stimulation intensity was 1 mA, and was maintained for 20 min. Each electrode pad was humidified with a 0.9% NaCl solution.

#### 2.5.2. Repetitive Transcranial Magnetic Stimulation (rTMS)

rTMS was used in patients older than 10 years and 11 months (MagStim Rapid^2^, Whitland, UK). A butterfly coil with air cooling system (MagStim Double 70 mm Air Film Coil) was used. The center of the coil was located over F3, back handed 45° from the midline, and a total of 1500 pulses were delivered in each session, at 1 Hz of frequency and an intensity of 90% of the resting motor threshold (i.e., the minimum stimulation intensity required to elicit a discernible hand muscle response in at least three of five consecutive pulses) [24]. Sessions were subdivided into four trains of 375 pulses, with a 1-min interval between each allowing for free movement of the head and neck. Most of the patients rejected the use of earplugs, and in the best-case scenario they agreed to listen to music by wearing headphones from their own devices.

### 2.6. Ethical Considerations

All the procedures followed the rules of the Declaration of Helsinki of 1975 for human research, and the study was approved by the scientific and ethics committee from the International Center for Neurological Restoration (CIREN37/2015). Parents gave written informed consent for their children to be considered for inclusion in the study, and where possible children also gave their consent.

## 3. Results

### 3.1. Change in Clinical Scales One Month after the Intervention

As a global result (G1 + G2), a significant decrease in the total score was observed in ADI-R, ABC, and ATEC scales one month after the intervention (Wilcoxon matched pair test; ABC, Z = 3.823, *p* = 0.000131; ADI-R, 3.337, *p* = 0.000846; ATEC, Z = 3.723, *p* = 0.000196). Figure 1a represents the global scores for the three main clinical scales we applied for the evaluation of autism behavior. The values are in correspondence with qualitative changes described by their relative uniqueness in socialization and communication domains, and with the GCIS (pre-intervention: 3.47 ± 0.6; post-intervention: 2.95 ± 0.2. Wilcoxon pair series test: Z = 2.803, *p* = 0.005062). All of the patients’ autistic behavior improved according to the scale results observed one month after the intervention. A comparison between the change in the total score of clinical scales did not show any significant differences correlated with the use of tDCS and rTMS (Mann–Whitney *U* Test, *p* > 0.05). Initial evaluation did not show any differences in the total score between groups when we looked for any evidence of age-dependent characteristics in both groups. The clinical response to NIBS was apparently independent of group age and type of intervention (See Figure 1b).

Figure 2 shows that no significant differences were observed between G1 and G2 in the initial clinical scale scores before the intervention (Mann–Whitney *U* test, *p* > 0.05); but values observed in G1 after NIBS showed a very important change when compared with the initial values (Wilcoxon; ABC, Z = 3.823, *p* = 0.000132; ADI-R, Z = 3.550, *p* = 0.000385 ;ATEC, Z = 5.526, *p* = 0.000241); the same behavior was seen in G2 after the intervention (Wilcoxon; ABC, Z = 3.295, *p* = 0.000982; ADI-R, Z = 3.588, *p* = 0.009633; ATEC, Z = 3.179, *p* = 0.001474).

### 3.2. Change in Clinical Scales during the First Six Months after the Intervention

Post-intervention follow-up was extended over six months. For both groups G1 and G2, NIBS induced a significant change in clinical scale scores (See Figure 2), and both groups maintained approximately the same values until the sixth month after NIBS, when there was a slight tendency for the total score in ABC, ADI-R, and ATEC to increase, but their punctuation remained lower than the observed values before treatment (See Figure 2). According to their parents’ opinion, stereotypes intensity and variety increased in frequency and number in many patients around the sixth month, but in clinical scales there were not any significant changes in specific domains.

Group 1 had lower values for all the scales one month after NIBS (solid lines); the same happened with G2 patients running in parallel, but only since the fourth month and ahead of follow-up (also one month after NIBS; dashed lines).

### 3.3. EEG-Based Brain Functional Connectivity Analysis

An increase in brain functional connectivity was observed after NIBS for patients who received either tDCS or rTMS, especially for higher frequencies: α, β, and γ bands showed the greatest increase, with a large increment in the number of 10/20 system points functionally related. The analysis of the effect in functional connectivity showed that the group that received rTMS accounted for the most significant changes when initial and final values were compared (see Figure 3).

### 3.4. ERP Analysis

Recordings were possible in only six patients in the initial evaluation; thus, after the intervention, only activity in those patients who collaborated during the task was recorded. All patients had a normal auditory brainstem response according to the normative laboratory data, but one patient had no ERP response. The P300 component showed a scalp distribution in the frontal and central regions, maximal in Fz. There was a shortening of P300 latency after treatment, with statistically significant differences with respect to the basal response (Wilcoxon matched pairs test, *p* = 0.043). P300 amplitude did not show statistically significant differences after treatment, even though the average value in the group was higher after the intervention (see Figure 4). The prolonged latency and low amplitude may be in agreement with the abnormal connectivity of the frontal lobes seen in ASD subjects [25], and engaged with attention circuits. The improvement of these parameters after NIBS might be a consequence of the improvement in functional connectivity and probably correlated with positive clinical changes in attentiveness, concentration, and speed of answering questions from a parent.

## 4. Discussion

Our results reproduce similar neuromodulatory effects to those reported previously by two other research groups who used the same target (L-DLPFC), applying either rTMS or tDCS [9,11,26]. Casanova et al. pioneered the use of therapeutic NIBS in autism and described the effects of one weekly low-frequency rTMS (1 Hz) during 12 weeks in 25 ASD children and adolescents. In that study they reported a significant reduction in repetitive and restricted behavior patterns and irritability, but no changes in social awareness or hyperactivity. They also described an increase in the amount of gamma activity in the EEG, shortening of the P300 component of ERP, and an increase of amplitude in the group receiving the intervention [9]. They used 150 TMS pulses per session, one session per week, and 12 sessions in total. In our study, the number of stimuli per session and the total number of sessions were considerably higher; this is in correspondence with recommended practical aspects for conventional use of NIBS in other diseases such as depression, obsessive compulsive disorder, chronic pain, etc., in order to obtain a sustained effect [27,28]. This could likely explain the significant clinical improvement in our group, not only in the abovementioned domains, but also in social aspects and communication. We also were able to establish that NIBS’ effects persisted into the sixth month after completion of treatment, a result that, as far as we know, has never been reported by any other group; this is a very important finding because it would probably indicate that patients would need a new treatment cycle (e.g., a second cycle of NIBS) six months after the previous intervention to maintain clinical improvement. Of course, it would also be interesting to know how the behavior of our patients will develop over the next six months and the forthcoming years (long-term outcome).

Another interesting double-blind and randomized trial in adult ASD patients was published by Enticott et al. using the dorsomedial prefrontal cortex as the target, and stimulating with the HAUT coil, which makes it possible to reach deep structures in the brain [12]. They applied 10 daily sessions of 5 Hz, 1500 pulses at resting motor threshold intensity. The clinical response was evaluated immediately after the intervention and one month later, and they reported a significant reduction in social relating symptoms (relative to sham participants) for both post-treatment assessments. Those in the active condition also showed a reduction in self-oriented anxiety during difficult and emotional social situations from pre-treatment to one-month follow-up. They suggested the need to use extended protocols such as those used to treat depression, and we agree that it is essential to achieving lasting effects for both NIBS methods, rTMS or tDCS [28].

D’Urso et al. published the results of an open pilot study using cathodic tDCS over the L-DLPFC in 10 young adults (18–25 years old) with 10 daily sessions of 1.5 mA, with evaluation two weeks after the intervention. They reported an improvement in autistic behavior of 26.7% according to the change observed in the clinical scale they applied; only two patients did not change their initial scale values, while the rest of the group improved in their symptoms [11]. There are two main differences in the tDCS protocol we applied in younger children (5–10 years old): first, again we used a higher number of sessions and second, the stimulation intensity in our case was 1 mA. NIBS in children seems to be as safe as in the adult population, so the same safety guidelines are indicated for its use in any population [29,30]. It is important to mention that in the case of tDCS, there is not enough evidence about the safety of 2 mA in young people (<18 years old), and most research has been done using 1 mA; however, there are also a few papers describing the absence of adverse effects with the use of 2 mA in pediatric patients [13].

In another interesting article, Amatachaya et al. described the effects of anodal tDCS over L-DLPF, with improvement in autistic behavior in 20 children with ASD. They designed a randomized, double-blind crossover and sham-controlled trial, applying 1 mA anodal tDCS over five consecutive days, and a washout period of four weeks This result reactivates the essential research question as to whether the clinical, biochemical, and physiological effects are polarity-dependent or not [10,31,32,33,34,35].

There are results from clinical trials with other diseases and physiological studies in which the opposite effects of cathodic and anodic tDCS have been well documented [35,36]. Our results using cathodic tDCS or low-frequency rTMS over the L-DLPFC share the same theoretical basis developed by Casanova et al. and D’Urso et al., though we studied a low sample number (24 patients), and by design we employed an open trial without the use of placebo stimulation. Nevertheless, the results are reproducible in G1 and G2, with patients being their own controls from the start of the intervention. We also had the opportunity to compare the effects of low-frequency rTMS and cathodic tDCS at the same time, but in ASD patients of different ages and with our experimental design we demonstrated that there are no differences, at least not in their clinical effects. Certainly, there was an important effect achieved by increasing functional connectivity with the use of both the stimulating methods, rTMS and tDCS, with a clear superiority of rTMS-induced changes. Of course, more research is needed to draw well-documented conclusions. Other authors have described improvements in particular aspects of adult ASD patients such as working memory and better syntax acquisition following single-dose anodal tDCS over the left and right dorsolateral prefrontal cortex (first case) or exclusively over the left dorsolateral prefrontal cortex [13,37] There were also positive changes in P300 ERP, but due to the low number of subjects in whom it was possible to carry out the passive oddball paradigm, we cannot make any comparison between the two different stimulating methods. Unfortunately, sample sizes for EEG-based connectivity analysis and ERP paradigm could not include all the patients, because, in close relationship with the complexity of the used method, when we carried out their initial electrophysiological evaluation (before intervention), not all recordings had a high enough quality for analysis.

## 5. Conclusions

Twenty sessions of NIBS over L-DLPFC improves autistic symptoms in ASD children, with a lasting effect of six months.

## Figures and Tables

**Figure 1 behavsci-07-00063-f001:**
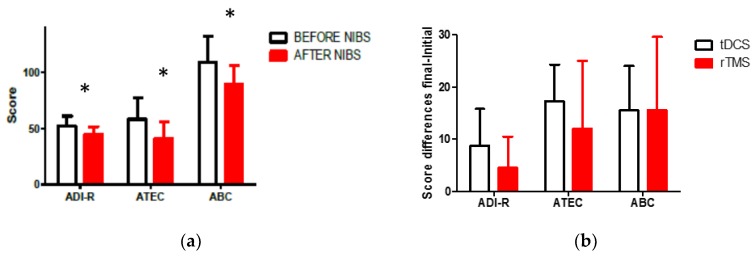
(**a**) Comparison of pre- intervention and one-month post-intervention scores in clinical scales; (**b**) no significant differences in the clinical effect with the use of rTMS or tDCS according to the score differences (Initial–Final) (* *p* < 0.05).

**Figure 2 behavsci-07-00063-f002:**
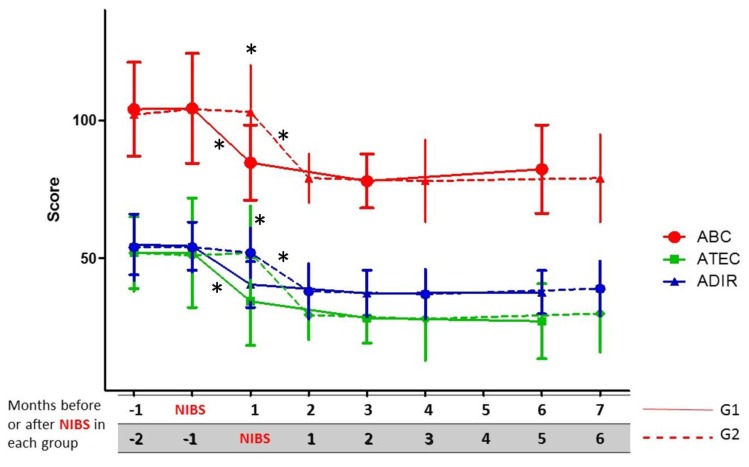
Short-term outcome based on clinical scale scores until six months of follow-up after NIBS in group 1 (G1) and group 2 (G2). * *p* < 0.05 (significant differences; comparisons between and within each group with scale scores before and after NIBS).

**Figure 3 behavsci-07-00063-f003:**
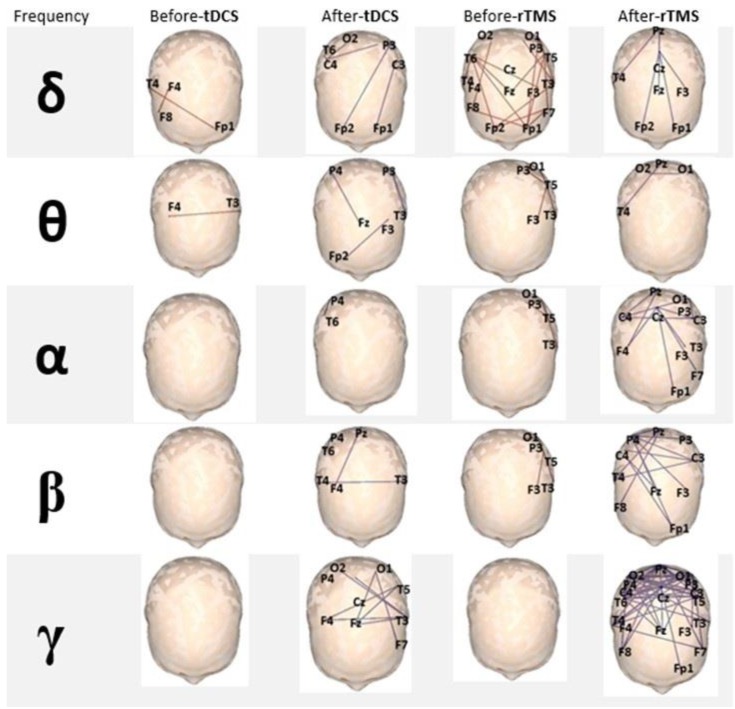
EEG-based analysis of brain functional connectivity. Note the positive change in all frequency bands after the intervention, but mainly in the γ band for both tDCS and rTMS, especially in the group that received rTMS (all electrodes’ locations are statistically significant, *p* < 0.05).

**Figure 4 behavsci-07-00063-f004:**
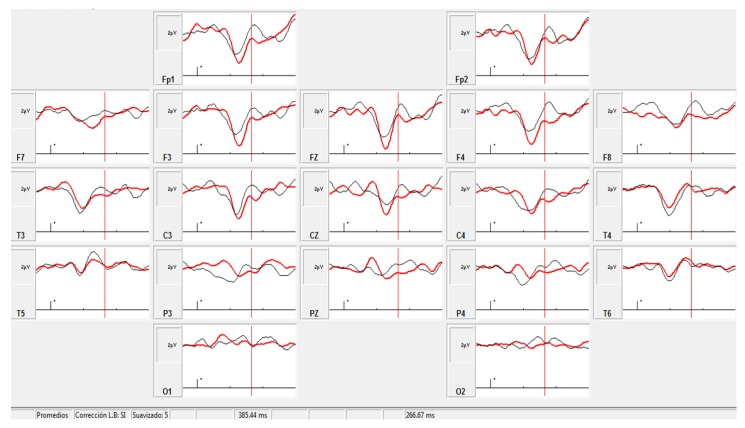
Grand averaged infrequent target. Black trace: pre-intervention (Amplitude and latency at Fz: 3.09 μV, 410 ms); red trace: post-intervention (Amplitude and latency at Fz: 3.28 μV, 371 ms).

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
