# Peer review of "Non-Invasive Brain Stimulation for Children with Autism Spectrum Disorders: A Short-Term Outcome Study"

_behavsci, 2017, doi:10.3390/bs7030063_

Round 1
Reviewer 1 Report
The manuscript: Non Invasive Brain Stimulation for children with Autism Spectrum Disorders. A short term outcome study, by Lazaro Gomez is a very nice study and has been written very well. I liked all the details of the material and methods. The differences between this study and Casanova's study is clear. Improvement in social aspects and communication of the treated patients are encouraging. I have two suggestion that might be beneficial:
1- Although this study reports the NIBS effects 6 months after the treatment, it would be nice to have some data about long term effects of such/similar stimulations on brain in general at least from studies by others if they are available.
3- It might be good to talk/predict a little bit more about the molecular mechanisms involved following such stimulations. There was a little about GABA in the introduction. Anything more might be helpful as well.
Author Response
Reviewer 1
Although this study reports the NIBS effects 6 months after the treatment, it
would be nice to have some data about long term effects of such/similar
stimulations on brain in general at least from studies by others if they are
available.
At least in ASD patients we have not found any other report of short term outcome, understanding that in general “short term” deals with evolution from the intervention to 1 year after. As far as we know there are at least other study with 3 months of follow-up, not yet published, but with a different methodology(personal communication) that we preferred not to include in our references. In other diseases, for example depression, there are intermediate and acceptable long term studies; but as we stress in the discussion: as far as we know, it is the first study describing outcome 6 months after NIBS in ASD patients.
It might be good to talk/predict a little bit more about the molecular
mechanisms involved following such stimulations. There was a little about
GABA in the introduction. Anything more might be helpful as well.
Yes, it is a good point; and we included some comments on GABAergic system in the first, second and third paragraph from the Introduction section.
Thank you so much for your comments!
Reviewer 2 Report
Review of manuscript by Gomez et al. titled “ Non Invasive Brain Stimulation for children with Autism Spectrum Disorders. A short term outcome study” submitted to Behavioral Sciences.
The paper describes application of non-invasive brain stimulation (NIBS), namely tDCS and rTMS in children with Autism Spectrum Disorder (ASD). Totally 24 children with confirmed ASD diagnosis were enrolled, and from those 15 (>10 yrs old) received 20 sessions of rTMS, while younger 9 subjects received 20 sessions of tDCS treatment. There were noted improvements in clinical evaluation scores (ADI-R, ABC, ATEC), at 1 month and 6 months follow-ups and improvements in EEG measures of functional connectivity (especially in higher frequency bands, i.e, gamma and beta). Six subjects were tested in an auditory oddball task and showed trend to P300 measures improvement. The study has relatively detailed review of rTMS and tDCS in treatment of ASD, though several more recent studies would be useful to add too. Interpretation of results goes along the concepts of increased cortical excitation/inhibition balance, decreased functional connectivity and other theoretical constructs such as “minicolumnar” neuropathology theory. Rationale for the application of NIBS therefore seems to be well justified based on above theoretical consideration. The most important result of the study is demonstration that NIBS effects may last up to 6 months, even though showed some trend to rebound back. Still this makes the study one of few with such extended follow-up, and it is not clear why the authors included “A short term outcome study” in the title, probably it is not needed. In general, the study is well written and has internal integrity.
There are only few minor issues that could be easily resolved and revised. The last paragraph at page 4 (lines 87-89) should be rewritten, probably by posing the hypothesis that 20 sessions of NIBS are expected to have positive neuromodulatory effect that may last for 6 months, rather than mentioning results. Results should not be yet mentioned in the Introduction section. In clinical evaluation instruments description there should be mentioned whether ABC and ATEC were done by parents, and what particular scores were used (probably T-scores). There is question whether ADI-R is good for relatively short-term outcome assessments, as most of clinicians consider this instrument as a diagnostic one. On Page 6 line 137 word “like” is excessive, not needed. Why only P300 was investigated in auditory oddball, it would be logical to look at Mismatch Negativity (MMN) as well, though P300 is also can be recorded and analyzed in such paradigm, so it is acceptable. It is not clear from those 6 children who complied with ERP test how many were from G1 and G2 groups (rTMS or tDCS). For EEG connectivity study it was described belonging of children to the group. It is somewhat disappointing that clinical results, EEG and ERP results are reported for different sample sizes, though results still deserve attention especially considering long-term outcome assessment. In the Results section (for example page 9, line 228) it is not necessary to interpret possible effects (placebo), this better be done in the Discussion section. Figures 1 mention *p<0.05, though there are no * on the figure, while on page 8 positive statistically significant outcomes for these measures are listed. On Figure 4 it is not clear for which topography (Fz or Cz etc) values of P300 amplitude are listed. In the methods description it is not clearly stated what rTMS frequency was used ( 1 Hz or 5 Hs or 0.5 Hz). The study was conducted without control group but still represents an important pilot investigation, especially considering that only few studies using NIBS were done in childrenThere are some technicality related suggestions regarding references, for example: In-text references: Line 51 Instead of [1;4]. Should be [1,4], The same: Line: 66, 79, 85, 262, 272, 304, 295, 306, References list does not fit the recommended format (ACS style); for example, Instead of: Casanova MF, Buxhoeveden DP, Switala AE, Roy E: Minicolumnar pathology in 338 autism. Neurology 2002;58:428-432.
Should be: Casanova, M.F.; Buxhoeveden, D.P.; Switala, A.E.; Roy, E. Minicolumnar pathology in 338 autism. Neurology 2002;58, 428-432.

Author Response
The study has relatively detailed review of rTMS and tDCS in treatment of ASD, though several more recent studies would be useful to add too.
Another reference from a recent study in ASD patients was included in the discussion, but we tried to use mainly studies from group of patients and at least 5 treatment sessions, looking for similarities in mehtodology.
Still this makes the study one of few with such extended follow-up, and it is not clear why the authors included “A short term outcome study” in the title, probably it is not needed.
It is our intention to stress that clinical effects last at least 6 months, in this way our study is different from the other papers that have been published(mostly 1week-2 months of follow up). For planning new treatment sessions in the future it is so important to argue how lasting the effects will be, and in medicine, we can say usually that long term effects are those effects lasting or appearing around 10 years ahead from any intervention or treatment, and short term effects 1 year or less after the intervention. From my point of view the term “short term outcome study” includes in the title an important information.
The last paragraph at page 4 (lines 87-89) should be rewritten, probably by posing the hypothesis that 20 sessions of NIBS are expected to have positive neuromodulatory effect that may last for 6 months, rather than mentioning results.
You are right. Corrected as the reviewer suggested.
Results should not be yet mentioned in the Introduction section.
You are right; the expressions were eliminated from the results.
In clinical evaluation instruments description, there should be mentioned whether ABC and ATEC were done by parents, and what particular scores were used (probably T-scores).
Yes, it was mentioned as the reviewer suggested.
There is question whether ADI-R is good for relatively short-term outcome assessments, as most of clinicians consider this instrument as a diagnostic one.
ADI-R is a very good diagnostic and time-consuming instrument for ASD, it has two qualification algorithms: 1. for diagnostic application; and 2. For current state characterization; this second was the one we applied for short-term outcome assessment and we think it is helpful, according to our results; also ATEC and ABC probably had a higher sensibility. In our study it was helpful.
On Page 6 line 137 word “like” is excessive, not needed.
Right. It was eliminated
Why only P300 was investigated in auditory oddball, it would be logical to look at Mismatch Negativity (MMN) as well, though P300 is also can be recorded and analyzed in such paradigm, so it is acceptable.
It is a very good point; in an earlier pilot study in 5 ASD child we started applying a MMN paradigm but results were disappointing, and then we tried just the most basic passive odd ball paradigm for P300, and it was possible in some cases to obtain acceptable quality recordings for analysis in the first electrophysiological evaluation(before intervention), but as you very well said:
It is somewhat disappointing that clinical results, EEG and ERP results are reported for different sample sizes, though results still deserve attention especially considering long-term outcome assessment.
Not in all patients that we recorded the EEG, it was possible to obtain acceptable quality artifact free segments for connectivity analysis; the incidence of movement artifacts was hugh, due to movements and poor collaboration in the firs electrophysiological evaluation(before intervention); so the answer to the two anterior comments are related.
It is not clear from those 6 children who complied with ERP test how many were from G1 and G2 groups (rTMS or tDCS). For EEG connectivity study it was described belonging of children to the group.
You are right! It was described as you suggested.
In the Results section (for example page 9, line 228) it is not necessary to interpret possible effects (placebo), this better be done in the Discussion section.
Correct. This comment was eliminated form the Result section.
Figures 1 mention *p<0.05, though there are no * on the figure, while on page 8 positive statistically significant outcomes for these measures are listed.
Right. It was added * to the significant values in Fig. 1
On Figure 4 it is not clear for which topography (Fz or Cz etc) values of P300 amplitude are listed.
Right. We added the specification of Fz electrode position in the figure and also in the methodology section.
In the methods description it is not clearly stated what rTMS frequency was used ( 1 Hz or 5 Hs or 0.5 Hz).
Right. It was 1 Hz, already declared in methodology.
The study was conducted without control group but still represents an important pilot investigation, especially considering that only few studies using NIBS were done in children.
Thank you!
There are some technicality related suggestions regarding references, for example: In-text references: Line 51 Instead of [1;4]. Should be [1,4], The same: Line: 66, 79, 85, 262, 272, 304, 295, 306, References list does not fit the recommended format (ACS style); for example, Instead of: Casanova MF, Buxhoeveden DP, Switala AE, Roy E: Minicolumnar pathology in 338 autism. Neurology 2002;58:428-432.
Should be: Casanova, M.F.; Buxhoeveden, D.P.; Switala, A.E.; Roy, E. Minicolumnar pathology in 338 autism. Neurology 2002;58, 428-432.
They were all arranged according to the journal format.
Thank you so much for your comments!